# A comprehensive DNA barcode inventory of Austria's fish species

**Lukas Zangl**[1,2]*, **Sylvia Schäffer**[1], **Daniel Daill**[1,3], **Thomas Friedrich**[4], **Wolfgang Gessl**[1], **Marija Mladinić**[5], **Christian Sturmbauer**[1], **Josef Wanzenböck**[6], **Steven J. Weiss**[1], **Stephan Koblmüller**[1]

1 Institute of Biology, University of Graz, Graz, Austria, 2 Universalmuseum Joanneum, Studienzentrum Naturkunde, Graz, Austria, 3 Consultants in Aquatic Ecology and Engineering—blattfisch e.U., Wels, Austria, 4 Institute of Hydrobiology and Aquatic Ecosystem Management, University of Natural Resources and Life Sciences, Vienna, Austria, 5 Department of Biology, Faculty of Science, University of Zagreb, Zagreb, Croatia, 6 Research Department for Limnology, Mondsee, University of Innsbruck, Mondsee, Austria

* lukas.zangl@uni-graz.at

**Data Availability Statement:** The data underlying this study can be found at BOLD (www.boldsystems.org) using the link: https://www.boldsystems.org/index.php/MAS_Management_DataConsole?codes=DS-AFISH (dx.doi.org/10.

## Abstract

Austria is inhabited by more than 80 species of native and non-native freshwater fishes. Despite considerable knowledge about Austrian fish species, the latest Red List of threatened species dates back 15 years and a systematic genetic inventory of Austria's fish species does not exist. To fulfill this deficit, we employed DNA barcoding to generate an up-to-date and comprehensive genetic reference database for Austrian fish species. In total, 639 newly generated cytochrome c oxidase subunit 1 (*COI*) sequences were added to the 377 existing records from the BOLD data base, to compile a near complete reference dataset. Standard sequence similarity analyses resulted in 83 distinct clusters almost perfectly reflecting the expected number of species in Austria. Mean intraspecific distances of 0.22% were significantly lower than distances to closest relatives, resulting in a pronounced barcoding gap and unique Barcode Index Numbers (BINs) for most of the species. Four cases of BIN sharing were detected, pointing to hybridization and/or recent divergence, whereas in *Phoxinus* spp., *Gobio* spp. and *Barbatula barbatula* intraspecific splits, multiple BINs and consequently cryptic diversity were observed. The overall high identification success and clear genetic separation of most of the species confirms the applicability and accuracy of genetic methods for bio-surveillance. Furthermore, the new DNA barcoding data pinpoints cases of taxonomic uncertainty, which need to be addressed in further detail, to more precisely assort genetic lineages and their local distribution ranges in a new National Red-List.

## Introduction

DNA barcoding was introduced as a suitable method for biological species discrimination in animals in 2003 [1], and since then the method has continued to receive unprecedented attention. For most animal groups, the region near the 5'-end of the cytochrome C oxidase subunit 1 (*COI*) is established as the standard barcoding marker. Despite certain valid reservations [e.g. 2–4], an enormous number of studies on various taxonomic groups (e.g., see [5] for plants

5883/DS-AFISH). In addition, data was uploaded to GenBank (ON097269 - ON097906).

**Funding:** Financial support was provided by the Austrian Federal Ministry of Science, Research and Economy in the frame of the ABOL (Austrian Barcode of Life; www.abol.ac.at) pilot project on vertebrates and an ABOL associated project within the framework of the "Hochschulraum-Strukturmittel" Funds. Financial support for covering the open access publication charges were covered by the University of Graz. DD is employed by a commercial company: Consultants in Aquatic Ecology and Engineering, Austria. We note that he contributed most of his work during the time of his Masters' thesis at the University of Graz and got employed by this company only recently. This company provided support in form of salary for him, but did not have any additional role in the study design, data collection and analysis, decision to publish, or preparation of the manuscript. The specific roles of all authors are articulated in the 'author contributions' section. The funders had no role in study design, data collection and analysis, decision to publish, or preparation of the manuscript.

**Competing interests:** DD is employed by a commercial company: Consultants in Aquatic Ecology and Engineering, Austria. There are no patents, products in development or marketed products associated with this research to declare. This does not alter our adherence to PLOS ONE policies on sharing data and materials.

[6], for insects [7,8], for amphibians and reptiles [9], for fungi, and [10] for fish) have accumulated over the last two decades. One particular upside of DNA barcoding is the breadth of useful applications. When applied to fishes, it can be used to investigate freshwater [10] or marine species [11–13], to determine species regardless of their ontogenetic stage [14–17] or to identify only residual parts of animals [18]. Furthermore, DNA barcoding data is increasingly used as a means for tracking catch records, food authenticity, mislabeling or fraud [19–22]. Moreover, freshwater ecosystems are among the most threatened throughout the world and freshwater species in Europe have experienced an 83% decline in populations over the last 50 years [23,24]. Habitat degradation, water pollution, river channel regulation, hydropower exploitation, invasive species and ultimately climate change entail a range of pressures that threaten freshwater biodiversity worldwide [24–26]. Furthermore, the high level of endemism within freshwater ecosystems, coupled with challenges in direct observation, requires tools for sound identification of species and evolutionary significant units to implement conservation efforts [27,28]. Species discrimination is also critical for biological monitoring and conservation purposes, hence DNA barcoding has gained additional importance in the light of recent alerts of biodiversity loss across all terrestrial and aquatic habitats [29,30]. Furthermore, biological surveillance increasingly encourages non-invasive sampling techniques like environmental DNA (eDNA) approaches [31,32], which heavily rely on high-quality genetic reference databases in order to facilitate reliable read identification and species assignment. Tracking biodiversity, however, requires precise species determination and while the identification of most adult (European) fishes can usually be achieved quite easily by experts, some morphologically challenging cases like the whitefishes (*Coregonus* spp.), minnows (*Phoxinus* spp.) or alien species like weatherfishes (*Misgurnus* spp.) [32–36] as well as the identification of juvenile fish remain difficult tasks [14–17]. In such cases, DNA barcoding might not necessarily replace classical morphology-based approaches as a stand-alone technique, but can aid as a complementary method to increase resolution [16,37,38]. However, in order to yield optimal identification results, DNA barcoding is heavily dependent on high quality, deep coverage reference libraries (e.g. the BOLD database [39]), which profit from the steady augmentation with unambiguously determined reference specimens [10]. Several national barcoding initiatives (such as GBOL, www.bolgermany.de; Barcoding Fauna Bavarica, barcoding-zsm.de/bfb; SWISSBOL, www.swissbol.ch; FINBOL, www.finbol.org; NORBOL, www.norbol.org) contribute their share and ensure continuity and the steady increase in reference data quality [40,41]. The Austrian Barcode of Life initiative (ABOL, www.abol.ac.at) is part of this international network aiming to contribute to this global database and, concomitantly investigate native biodiversity.

Based on the latest Austrian Red List of endangered teleost fish and lamprey species from 2007 [42] as well as other literature on the Austrian fish fauna [44], approximately 85 fish species are present in Austria, 70 of which are considered native. However, these literature sources differ widely concerning some taxa. For example, the genus *Coregonus* accounts for 12 out of 85 species in [42], but only a single entity in [43], where it was considered to be a "species complex" due to taxonomic uncertainties. As the current Red List was compiled almost 15 years ago (last version from 2007) and new/alien invaders/species/lineages [34,35,45–48] have been recently recorded, the current ABOL-project also provides a valuable source of data for an update of the current Red List of Austrian teleost fish and lampreys, and a timely overview of the current freshwater fish diversity of Austria. Comprehensive knowledge on fish diversity is key for designing appropriate conservation action plans and may also support initial assessment of the need for management actions to be taken against invasive species.

Taken together, this study aims to i) add unambiguously determined reference specimens of Austrian fish to the international barcode of life database (BOLD), ii) contribute to the current understanding of the Austrian fish fauna and investigate the extant diversity (loss of

species in the wild, new invaders/introductions) and iii) test the discriminating power of DNA barcoding for Austrian fishes.

## Material and methods

The cumulative combination of all teleost fish and lamprey species listed in [43,44] as well as the current Red List for Austrian freshwater fishes [42] was used to define the extant freshwater fish diversity in Austria. According to the literature, 70 out of 85 species are listed as native. Additionally, a newly described species of gudgeon [45] and an alien species of weatherfish [48] have been added to the known fish diversity. In order to comprehensively cover the Austrian species assemblage, the present dataset consists of two sources of barcode sequences: i) *COI* sequences of Austrian fish species already available from BOLD ([32,34,35,45–48] including unpublished records (iBOL data release)) and ii) new *COI* barcode sequences generated in the course of this study. At the time this dataset was compiled, 1,048 *COI* sequences of Austrian fishes were available on BOLD (22.03.2021). Of those, samples not identified to the species level as well as all samples with sequences less than 500 bp in length were excluded, leaving 377 BOLD sequences. For more in-depth analyses of potentially ambiguous taxa pinpointed by the initial investigation (see below), sequences from other regions of Europe, outside of Austria, were downloaded from BOLD and compiled into separate datasets for *Phoxinus* spp. ([10,32,34,35,45,49–58], DS-EPHO (dx.doi.org/10.5883/DS-EPHO)) and *Barbatula barbatula* ([10,49–51,59,60], DS-EBBAR (dx.doi.org/10.5883/DS-EBBAR)). For the fresh material, all samples were opportunistically obtained in the framework of licensed electrofishing surveys in the years 2014–2021 conducted by a variety of private and public authorities. The rest of the samples were donated by state natural history museums (Natural History Museum Vienna, Oberösterreichisches Landesmuseum Linz). All newly collected specimens are stored permanently at Natural History Museums [see project code 'BCAFL' on BOLD (www.boldsystems.org) for sampling and taxonomic information]. Fin clips were taken and stored in pure ethanol at -20°C. Extraction of DNA of all 689 samples from 70 localities (Fig 1) followed a rapid Chelex protocol [61]. PCR, chain termination sequencing and SephadexTM G-50 (Amersham Biosciences) purification of the DNA barcode region (*COI*) amplicons using the primer combination C_FishF1t1 and C_FishR1t1 [62] and FishF1 and Str_R [63] followed [64,65] with the BioTherm DNA polymerase (GeneCraft Germany) and 50°C annealing temperature being the only alterations. Sequences were visualized on an ABI 3500xl capillary sequencer (Applied Biosystems).

All sequences were edited manually using MEGA 6.06 [66] and uploaded to the BOLD database, and are accessible under the project 'ABOL–Barcoding of the Austrian fish and lampreys (BCAFL)'. The final dataset of both downloaded and newly generated sequences consisted of 1,016 sequences (DS-AFISH dx.doi.org/10.5883/DS-AFISH) for subsequent analyses (see Table 1 for number of sequences per species). Visualization of sequence similarity clustering was conducted using the 'Taxon ID Tree' tool implemented on BOLD with the BOLD aligner algorithm. Intra ($I_{max}$)- and interspecific genetic distances (distance to nearest neighbor–DNN) were calculated under the K2P model with the 'Barcode Gap Analysis' tool also implemented on BOLD (K2P distance model, BOLD aligner, complete deletion for ambiguous base/gap handling). Furthermore, both distance-based, Automatic Barcode Gap Discovery' (ABGD, [67]) or 'Assemble Species by Automatic Partitioning' (ASAP, [68]), and tree-based, the 'Bayesian Poisson Tree Processes' model (bPTP, [69]), species delimitation methods were conducted. For ABGD, the alignment containing all sequences was downloaded from BOLD and uploaded to the ABGD webserver (https://bioinfo.mnhn.fr/abi/public/abgd/abgdweb.html). Analyses were run with the Kimura (K2P) TS/TV model with the preset parameters

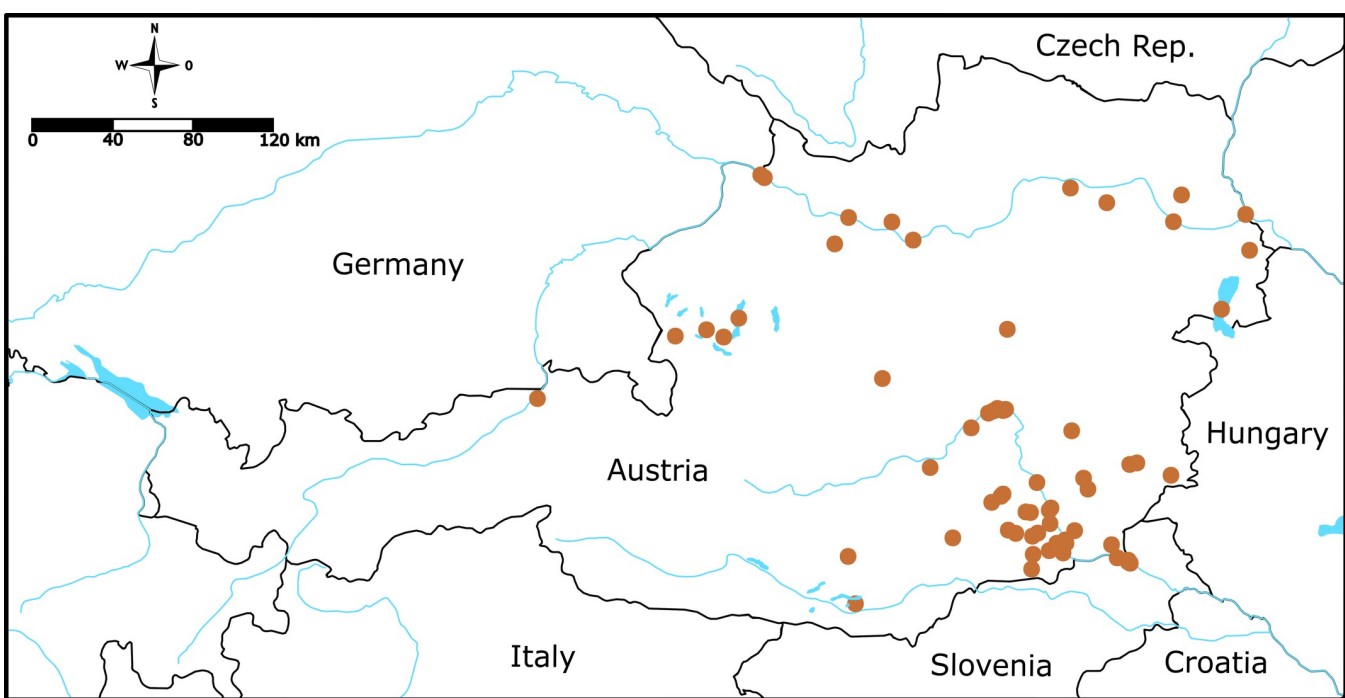

**Fig 1. Map of sampling localities.** Map of Austria and surrounding countries indicating the sampling locations of newly sequenced samples (orange spheres).

(Pmin: 0.001, Pmax: 0.1, Steps: 10, X (relative gap width): 1.5). The same procedure was conducted for ASAP, also run from a webserver (https://bioinfo.mnhn.fr/abi/public/asap/asapweb.html) with the default parameters. For the bPTP analysis, the phylogenetic input tree was inferred using the IQ-TREE webserver (http://iqtree.cibiv.univie.ac.at/) with the automatic substitution model and 1000 ultrafast bootstrap replicates [70]. The resulting tree was converted to Newick format in FigTree v1.4.4 (http://tree.bio.ed.ac.uk/software/figtree/) and uploaded to the bPTP webserver (https://species.h-its.org/ptp/) where the analysis was run with 100,000 MCMC generations, the thinning set to 100, a burn-in fraction of 0.1 and a random seed [69].

## Results

From the 689 samples covering all but one of the extant families (only Anguillidae is missing), 96% of the genera and 95% of all fish species present in Austria (based on [42–44]), 639 *COI* barcodes ranging from 512 to 700 bp in length were generated, representing an overall sequencing success rate of 93%. All sequences are accessible on BOLD (project code 'BCAFL') and GenBank (ON097269—ON097906). The overall dataset (1,016 sequences), including downloaded records from Austrian fish samples, covers a total of 94% of all families, 98% of all genera and 96% of all species present in Austria. The sequence similarity clustering resulted in 84 distinct clades largely mirroring morphological species identification and 83 Barcode Index Numbers (BINs, Fig 2).

One specimen originally identified as Prussian carp (*Carassius gibelio*) was quite divergent from other alleged *C. gibelio* samples. A BLAST search in BOLD/GenBank indicated, with 100% sequence similarity, that this divergent haplotype sampled in Schwarzaubach in Styria most likely represents the Ginbuna, *Carassius langsdorfii*, a species hitherto unknown for Austria. In addition to this new record, discordances between currently accepted species, DNA

**Table 1. K2P distances (in %) of *COI* sequences within and between Austrian fish species.**

| Species | BIN | N | $I_{max}$ | Nearest neighbor | DNN |
|---|---|---|---|---|---|
| Acipenseriformes | | | | | |
| Acipenseridae | | | | | |
| *Acipenser ruthenus* | BOLD:AAA8921 | 3 | 0 | *Huso huso* | 5.32 |
| *Acipenser stellatus* | BOLD:AAA3851 | 1 | na | *Huso huso* | 6.34 |
| *Huso huso* | BOLD:AAA3852 | 2 | 0 | *Acipenser ruthenus* | 5.32 |
| Centrarchiformes | | | | | |
| Centrarchidae | | | | | |
| *Lepomis gibbosus* | BOLD:AAA5641 | 14 | 0.3 | *Ctenopharyngodon idella* | 19.67 |
| Cypriniformes | | | | | |
| Acheilognathidae | | | | | |
| *Rhodeus amarus* | BOLD:AAC4093 | 15 | 0 | *Ballerus ballerus* | 17.06 |
| Cobitidae | | | | | |
| *Cobitis elongatoides* | BOLD:ACE4983 | 17 | 0.9 | *Misgurnus bipartitus* | 11.73 |
| *Misgurnus bipartitus* | BOLD:ACB5380 | 2 | 0 | *Cobitis elongatoides* | 11.73 |
| *Misgurnus fossilis* | BOLD:AAK6219 | 5 | 0.3 | *Sabanejewia balcanica* | 16.56 |
| *Sabanejewia balcania* | BOLD:AAE3193 | 13 | 0.9 | *Cobitis elongatoides* | 16.07 |
| Cyprinidae | | | | | |
| *Barbus balcanicus* | BOLD:AAC5468 | 11 | 0.3 | *Barbus barbus* | 4.68 |
| *Barbus barbus* | BOLD:AAD1959 | 29 | 0.3 | *Barbus balcanicus* | 4.68 |
| *Carassius auratus* | BOLD:AAA7176** | 1 | na | *Carassius gibelio* | 0 |
| *Carassius carassius* | BOLD:AAN9565 | 4 | 0 | *Carassius gibelio* | 7.64 |
| *Carassius gibelio* | BOLD:AAA7176** | 12 | 0.9 | *Carassius auratus* | 0 |
| *Carassius langsdorfii* | BOLD:AAA7176** | 1 | na | *Carassius gibelio* | 4.33 |
| *Cyprinus carpio* | BOLD:AAA7175 | 8 | 0.3 | *Carassius gibelio* | 9.59 |
| Gobionidae | | | | | |
| *Gobio spp.* | BOLD:AAC5607; BOLD:ADH1249; BOLD:ABY6890 | 62 | 3.69 | *Romanogobio carpathorossicus* | 12.72 |
| *Pseudorasbora parva* | BOLD:AAD0138 | 10 | 0.6 | *Romanogobio vladykovi* | 16.31 |
| *Romanogobio carpathorossicus* | BOLD:ABV4495 | 19 | 1.2 | *Romanogobio vladykovi* | 10.65 |
| *Romanogobio skywalkeri* | BOLD:ADH6027 | 27 | 0.3 | *Romanogobio uranoscopus* | 7.98 |
| *Romanogobio uranoscopus* | BOLD:AAF7823 | 9 | 0.9 | *Romanogobio vladykovi* | 5.28 |
| *Romanogobio vladykovi* | BOLD:AAC5609 | 36 | 0.9 | *Romanogobio uranoscopus* | 5.28 |
| Leuciscidae | | | | | |
| *Abramis brama* | BOLD:AAC8592* | 9 | 0.3 | *Blicca bjoerkna* | 0 |
| *Alburnoides bipunctatus* | BOLD:AAC4344 | 26 | 1.2 | *Ballerus sapa* | 9.98 |
| *Alburnus alburnus* | BOLD:AAB6906 | 35 | 0.9 | *Alburnus chalcoides* | 2.73 |
| *Alburnus chalcoides* | BOLD:AAB6908 | 9 | 0.6 | *Alburnus alburnus* | 2.73 |
| *Ballerus ballerus* | BOLD:AAZ6088 | 1 | na | *Ballerus sapa* | 2.13 |
| *Ballerus sapa* | BOLD:AAF3389 | 6 | 0 | *Ballerus ballerus* | 2.13 |
| *Blicca bjoerkna* | BOLD:AAD3588 | 7 | 4.68 | *Abramis brama* | 0 |
| *Chondrostoma nasus* | BOLD:AAD7920 | 40 | 1.2 | *Telestes souffia* | 5.6 |
| *Leucaspius delineatus* | BOLD:ACF4430 | 1 | na | *Alburnus alburnus* | 6.28 |
| *Leuciscus aspius* | BOLD:AAC8137 | 13 | 0.3 | *Leuciscus idus* | 5.33 |
| *Leuciscus idus* | BOLD:AAD5733 | 5 | 0 | *Leuciscus leuciscus* | 0 |
| *Leuciscus leuciscus* | BOLD:AAD5733 | 8 | 0.6 | *Leuciscus idus* | 0 |
| *Pelecus cultratus* | BOLD:AAF5575 | 4 | 0 | *Ballerus ballerus* | 10.72 |
| *Phoxinus lumaireul* | BOLD:AAC8034 | 19 | 2.43 | *Phoxinus phoxinus* | 0 |

*(Continued)*

**Table 1.** (Continued)

| Species | BIN | N | $I_{max}$ | Nearest neighbor | DNN |
|---|---|---|---|---|---|
| *Phoxinus phoxinus* | BOLD:AAC8034; BOLD:AAC8036; BOLD:ADL2661; BOLD:ACE5740 | 63 | 6.28 | *Phoxinus lumaireul* | 0 |
| *Rutilus meidingeri* | BOLD:AAA5494 | 9 | 0 | *Rutilus rutilus* | 4.3 |
| *Rutilus rutilus* | BOLD:ABZ3785 | 26 | 0.9 | *Rutilus meidingeri* | 4.3 |
| *Rutilus virgo* | BOLD:AAE3231; BOLD:ADG8651 | 5 | 2.11 | *Rutilus rutilus* | 5.59 |
| *Scardinius erythrophthalmus* | BOLD:AAC1452 | 16 | 1.81 | *Alburnus chalcoides* | 8.25 |
| *Squalius cephalus* | BOLD:AAD8346 | 36 | 1.81 | *Chondrostoma nasus* | 6.26 |
| *Telestes souffia* | BOLD:AAE9853 | 6 | 0.9 | *Chondrostoma nasus* | 5.6 |
| *Vimba vimba* | BOLD:AAD9149 | 8 | 0 | *Blicca bjoerkna* | 3.36 |
| Nemacheilidae | | | | | |
| *Barbatula barbatula* | BOLD:AAA1239; BOLD:AAA1243 | 17 | 4.66 | *Misgurnus fossilis* | 19.82 |
| Tincidae | | | | | |
| *Tinca tinca* | BOLD:AEJ6454 | 5 | 2.13 | *Hypophthalmichthys molitrix* | 8.88 |
| Xenocyprididae | | | | | |
| *Ctenopharyngodon idella* | BOLD:ACL1923 | 4 | 0 | *Hypophthalmichthys molitrix* | 9.75 |
| *Hypophthalmichthys molitrix* | BOLD:AAF6633 | 4 | 0.6 | *Hypophthalmichthys nobilis* | 4.98 |
| *Hypophthalmichthys nobilis* | BOLD:ADK6840 | 1 | na | *Hypoophthalmichthys molitrix* | 4.98 |
| Esociformes | | | | | |
| Esocidae | | | | | |
| *Esox lucius* | BOLD:AAA5988 | 9 | 0.3 | *Sander lucioperca* | 20.17 |
| Umbridae | | | | | |
| *Umbra krameri* | BOLD:AAO6269 | 2 | 0 | *Salmo trutta* | 17.11 |
| Gadiformes | | | | | |
| Gadidae | | | | | |
| *Lota lota* | BOLD:AAB2046 | 14 | 0.6 | *Huso huso* | 19.77 |
| Gobiiformes | | | | | |
| Gobiidae | | | | | |
| *Babka gymnotrachelus* | BOLD:AAX5968 | 1 | na | *Ponticola kesslerii* | 9.01 |
| *Neogobius melanostomus* | BOLD:AAC0218 | 28 | 0 | *Ponticola kesslerii* | 16.76 |
| *Ponticola kesslerii* | BOLD:AAD8740 | 9 | 0 | *Babka gymnoctrachelus* | 9.01 |
| *Proterorhinus semilunaris* | BOLD:AAD0669 | 11 | 0 | *Ponticola kesslerii* | 13.21 |
| Perciformes | | | | | |
| Cottidae | | | | | |
| *Cottus gobio* | BOLD:ABX6144 | 36 | 2.42 | *Pungitius pungitius* | 18.21 |
| Gasterosteidae | | | | | |
| *Gasterosteus aculeatus* | BOLD:AAA8488 | 12 | 0.9 | *Pungitius pungitius* | 17.62 |
| *Pungitius pungitius* | BOLD:AAA8317 | 6 | 0 | *Gasterosteus aculeatus* | 17.62 |
| Percidae | | | | | |
| *Gymnocephalus baloni* | BOLD:AAL5632 | 1 | na | *Gymnocephalus schraetser* | 2.75 |
| *Gymnocephalus cernua* | BOLD:ACO0744 | 7 | 0.3 | *Gymnocephalus schraetser* | 4.99 |
| *Gymnocephallus schraetser* | BOLD:AAB0394 | 6 | 0 | *Gymnocephalus baloni* | 2.75 |
| *Perca fluviatilis* | BOLD:AAB0356 | 28 | 1.2 | *Sander lucioperca* | 13.83 |
| *Sander lucioperca* | BOLD:AAD1749 | 11 | 0.9 | *Sander volgensis* | 4.0 |
| *Sander volgensis* | BOLD:AAJ5463 | 4 | 0 | *Sander lucioperca* | 4.0 |
| *Zingel streber* | BOLD:AAE6523 | 10 | 0.3 | *Zingel zingel* | 6.58 |
| *Zingel zingel* | BOLD:AAH8409 | 15 | 0 | *Zingel streber* | 6.58 |
| Petromyzontiformes | | | | | |

*(Continued)*

**Table 1.** (Continued)

| Species | BIN | N | $I_{max}$ | Nearest neighbor | DNN |
|---|---|---|---|---|---|
| Petromyzontidae | | | | | |
| *Eudontomyzon mariae* | BOLD:ABY5382 | 17 | 5.25 | *Lampetra planeri* | 4.0 |
| *Lampetra planeri* | BOLD:AAB6058 | 7 | 0 | *Eudontomyzon mariae* | 4.0 |
| Salmoniformes | | | | | |
| Salmonidae | | | | | |
| *Coregonus spp.* | BOLD:ACA5470 | 12 | 0 | *Salmo trutta* | 12.22 |
| *Hucho hucho* | BOLD:AAE1471 | 8 | 0 | *Salmo trutta* | 10.41 |
| *Oncorhynchus mykiss* | BOLD:AAA1627 | 7 | 0.6 | *Salvelinus umbla* | 9.63 |
| *Salmo salar* | BOLD:AAA3435 | 2 | 0 | *Salmo trutta* | 6.59 |
| *Salmo trutta* | BOLD:AAB3872 | 26 | 0.9 | *Salmo salar* | 6.59 |
| *Salvelinus fontinalis* | BOLD:AAC3575 | 5 | 0.6 | *Salvelinus umbla* | 7.6 |
| *Salvelinus umbla* | BOLD:ABZ0871 | 6 | 0.3 | *Salvelinus fontinalis* | 7.6 |
| *Thymallus thymallus* | BOLD:AAD6463 | 18 | 2.13 | *Coregonus spp.* | 14.26 |
| Siluriformes | | | | | |
| Ictaluridae | | | | | |
| *Ameiurus melas* | BOLD:AAA7255*** | 2 | 0 | *Ameiurus nebulosus* | 2.75 |
| *Ameiurus nebulosus* | BOLD:AAA7255*** | 6 | 0 | *Ameiurus melas* | 2.75 |
| Siluridae | | | | | |
| *Silurus glanis* | BOLD:ACL1933 | 5 | 0 | *Ameiurus melas* | 18.05 |

Barcode Index Numbers (BIN), the number sequences per species (N), the maximum intraspecific ($I_{max}$) and the minimum distance (DNN) to the nearest neighbor are given.

* indicates the cluster of the common bream (*Abramis brama*), which contains one sequence of a morphologically clearly determined *Blicca bjoerkna*.

**indicates the cluster of *Carassius gibelio*, *C. langsdorfii* and *C. auratus*, which share the same BIN, but appear on distinct branches on the NJ tree and can also clearly be determined based on their morphology.

*** indicates *Ameiurus nebulosus* and *A. melas* which share a BIN but result on distinct branches on the NJ tree. Note, the systematic classification used here is based on [71] except for gudgeons of the genus *Romanogobio*, where we follow [45], trouts of the genus *Salmo*, where we follow [72] and coregonids of the genus *Coregonus*, which cannot be distinguished by DNA barcodes due to recent diversification [10].

barcodes and BIN assignment were detected in gudgeons of the genus *Gobio* (three distinct clades, three individual BINs), minnows of the genus *Phoxinus* (four distinct clades, 4 individual BINs) and stone loaches (*Barbatula barbatula*, two distinct clades, two individual BINs). Additionally, two different BINs were detected in the Danube roach (*Rutilus virgo*, BOLD: AAE3231 and BOLD:ADG8651) including a unique new BIN for Austria. Furthermore, BIN sharing was detected in four cases (*Leuciscus leuciscus/L. idus*, BOLD:AAD5733; *Abramis brama/Blicca bjoerkna*, BOLD:AAC8592; *Carassius langsdorfii/C. gibelio/C. auratus*, BOLD: AAA7176 and *Ameiurus nebulosus/A. melas*, BOLD:AAA7255). These results were also largely reflected by the analysis of genetic distances (Table 1).

With mean intra- and interspecific distances of 0.22 and 6.49% respectively, the barcode gap (i.e., interspecific distances exceeding intraspecific distances) was well reflected for most of the species (Fig 3). Only *Blicca bjoerkna* (maximum intraspecific distance ($I_{max}$): 4.68 (due to a single morphologically clear *B. bjoerkna* specimen with introgressed *Abramis brama* mtDNA), the species/lineages of *Phoxinus* spp. ($I_{max}$: 6.28) and *Eudontomyzon mariae* ($I_{max}$: 5.25) showed higher intraspecific than interspecific distances. Additionally, distances to conspecifics exceeding 1.0% were also detected within *Alburnoides bipunctatus*, *Barbatula barbatula*, *Chondrostoma nasus*, *Cottus gobio*, *Gobio* spp., *Perca fluviatilis*, *Romanogobio carpathorossicus*,

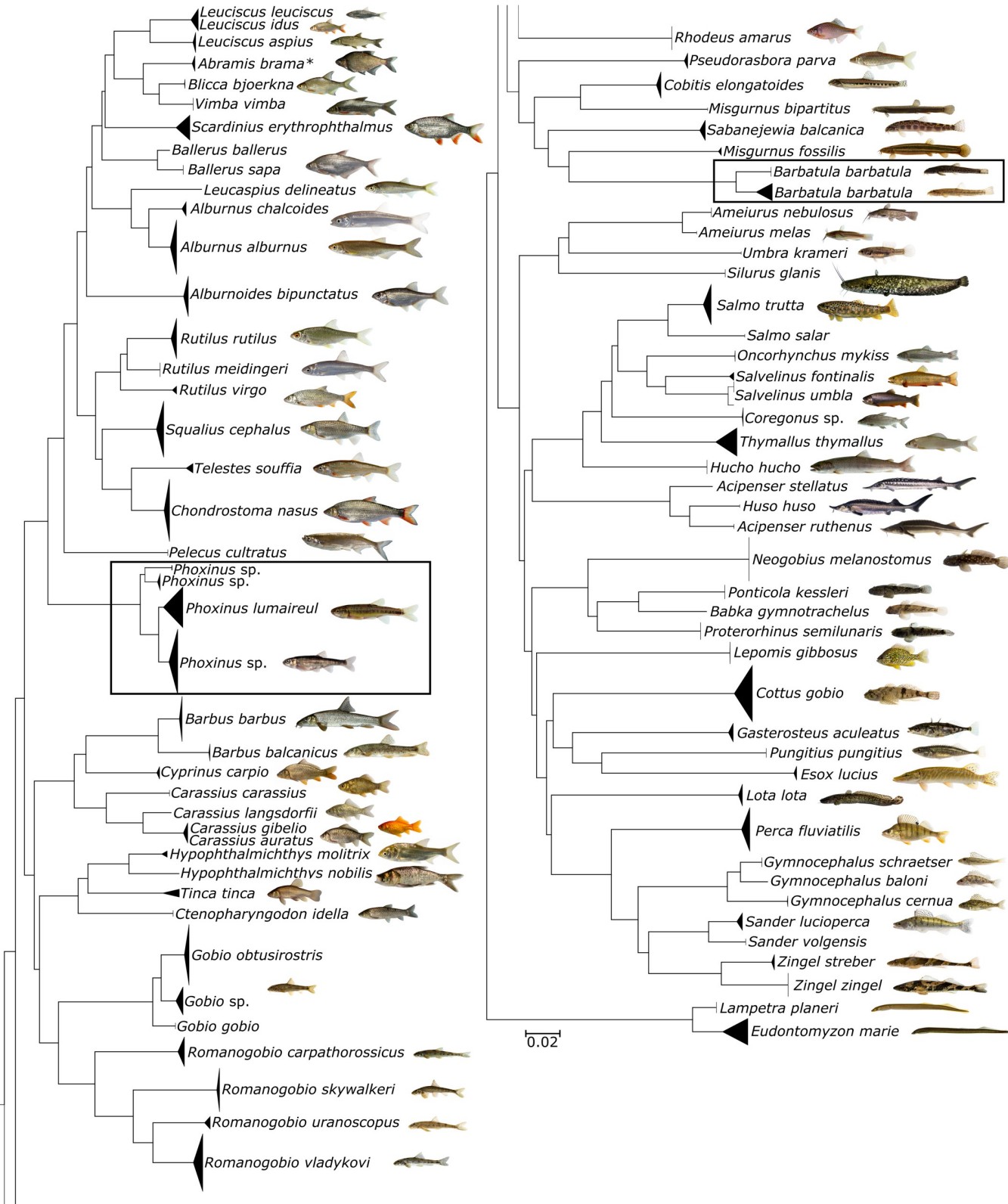

**Fig 2. NJ tree based on DNA barcode sequences of Austrian fish species.** NJ tree of Austrian teleost fish and lamprey species based on K2P distances of 1,016 *COI* DNA barcode sequences. The topology of the tree was inferred with the "Taxon ID Tree" tool implemented in BOLD and visualized in FigTree v1.4.4 (http://tree.bio.ed.ac.uk/software/figtree/). Black frames mark species that are investigated in a broader geographic context further below.

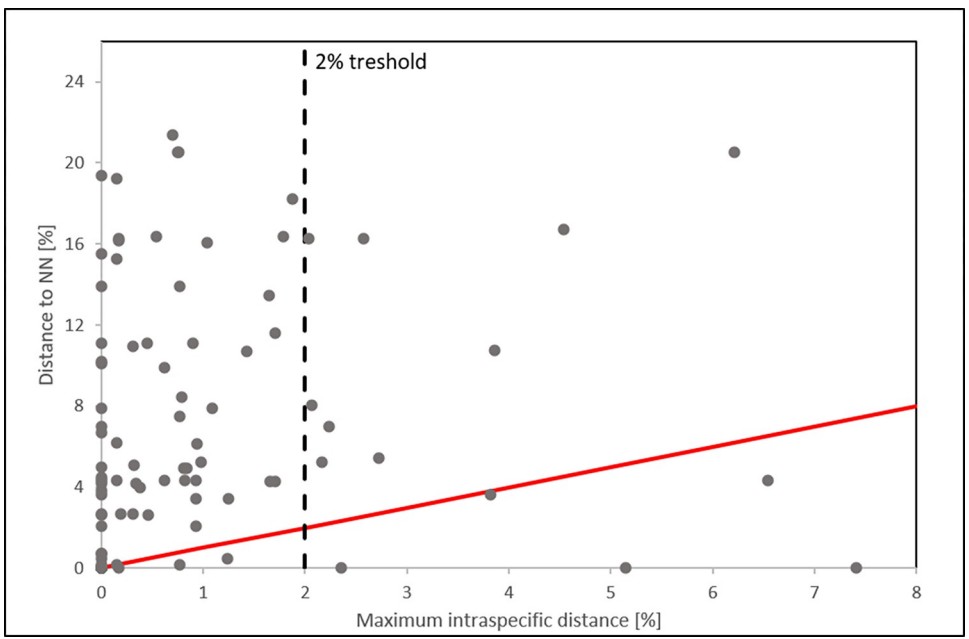

**Fig 3. Visualization of the 'Barcode gap'.** Barcode gap plot of the distance to the nearest neighbor (NN) vs. the maximum intraspecific distance of Austrian fish species. Dots above the red line suggest the presence of a barcoding gap. Outliers were detected in *Abramis brama*, *Barbus barbus*, *Blicca bjoerkna*, *Carassius auratus*, *Carassius gibelio*, *Eudontomyzon marie*, *Leuciscus idus*, *Leuciscus leuciscus* and *Phoxinus* spp.

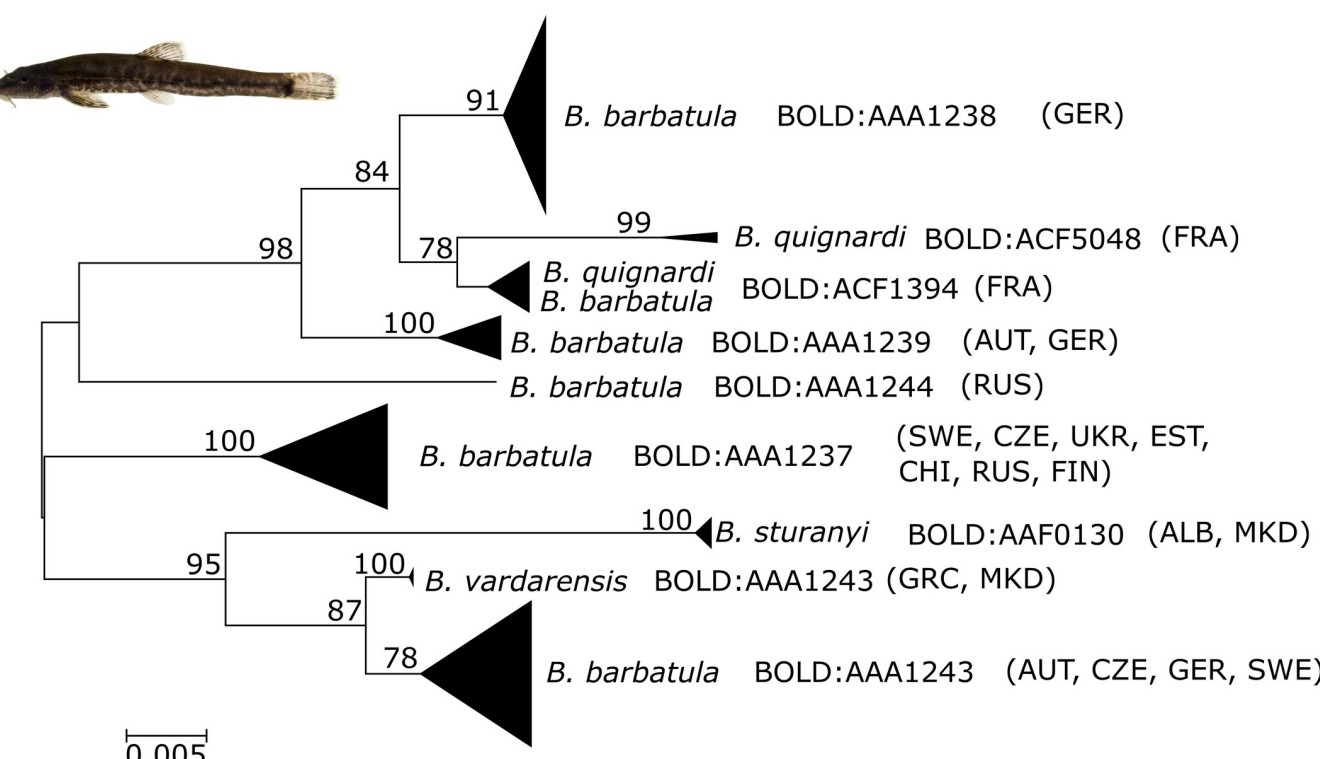

**Fig 4. NJ tree of European *Barbatula* species.** Phylogeny of European *Barbatula* species based on *COI* barcode sequences available on BOLD and from this study. Species names and BINs are given, countries of origin are indicated by acronyms in parentheses.

*Rutilus virgo*, *Scardinius erythrophthalmus*, *Squalius cephalus*, *Tinca tinca* and *Thymallus thymallus* (Table 1).

However, except for *Gobio* spp. and *Barbatula barbatula* these cases did not result in additional BINs. Similar results were also obtained from the other species delimitation analyses (see S1 Table). ABGD resulted in 88 species in the initial and 90 species in the recursive partition using a prior maximal distance of P = 0.0129. ASAP on the other hand reported 65–91 partitions/species based on the ten best partitioning schemes regarding the ASAP score. Even though the exact grouping of samples/species varies slightly between the individual priors and partitions, the overall patterns are the same, e.g., *Gobio* gudgeons are lumped into two groups despite the three lineages found by [47], *Phoxinus* minnows result in at least three distinct groups and that *Ameiurus nebulosus* and *A. melas* result in different groups despite their shared BIN. Finally, the maximum likelihood partitioning of the tree-based bPTP resulted in 88 species. Analysis of available pan-European stone loach data revealed at least five distinct lineages (and BINs) of *Barbatula barbatula* in Europe (Fig 4). Two of those lineages are solely comprised by samples from Germany or Russia, while the other three lineages contain samples from several countries reflecting a geographical pattern with an eastern (Germany and Austria), Danubian, and Northeastern European clade. Interestingly, the Danubian clade branches off from *B. vardarensis* native to Greece and North Macedonia, with which it shares a common BIN (BOLD:AAA1243).

Minnows of the genus *Phoxinus*, however, revealed a far more complex pattern based on their *COI* sequences. Besides country- and Balkan-specific MOTUs, four Central European lineages containing samples from several countries including Austria were found (Fig 5, S1 Table). Assignment of species names to these molecular taxonomic units (MOTUs) proved difficult, as each cluster contained specimens of various determinations (e.g. BOLD:ADL2661 contained *Phoxinus* sp., *P. phoxinus* and *P. marsilii*). Nonetheless, our results are wholly congruent with the presence of more than one species of *Phoxinus* in Austria and consequently also in Europe [35].

## Discussion

In this study, we present an almost complete DNA barcode reference inventory for Austrian fishes. From the 639 newly generated *COI* barcode sequences, only the European eel (*Anguilla anguilla*) as well as two sturgeon species, namely the Russian sturgeon (*Acipenser gueldenstaedtii*) and the ship sturgeon (*Acipenser nudiventris*), which have also been listed for Austria [43], are missing. For the two former species, PCRs (of old museum tissue) were unsuccessful, for the latter species no samples could be obtained. For all species, two or more samples were obtained, except for the racer goby (*Babka gymnotrachelus*), Balon's ruffe (*Gymnocephalus baloni*), the stellate sturgeon (*Acipenser stellatus*), the blue bream (*Ballerus ballerus*), the sunbleak (*Leucaspius delineatus*) and the bighead carp (*Hypophthalmichthys nobilis*), for which only a single sample was available. Whitefish (*Coregonus* spp.) were not treated as distinct species in our study as there is no consensus yet on whether the different forms found in the different lakes represent different species or ecotypes and because previous studies have shown that divergence of these species/ecotypes is too recent to be fully resolved by mtDNA data [73,74]. These issues are further complicated by hybridization with closely related introduced species throughout their ranges [73,74]. Similar to previous studies [10], analysis of the DNA barcoding data largely mirrors the known national species inventory. However, we found a few cases of BIN sharing and deep intraspecific divergence, potentially indicating cryptic diversity and/or new species records for Austria, in our new dataset.

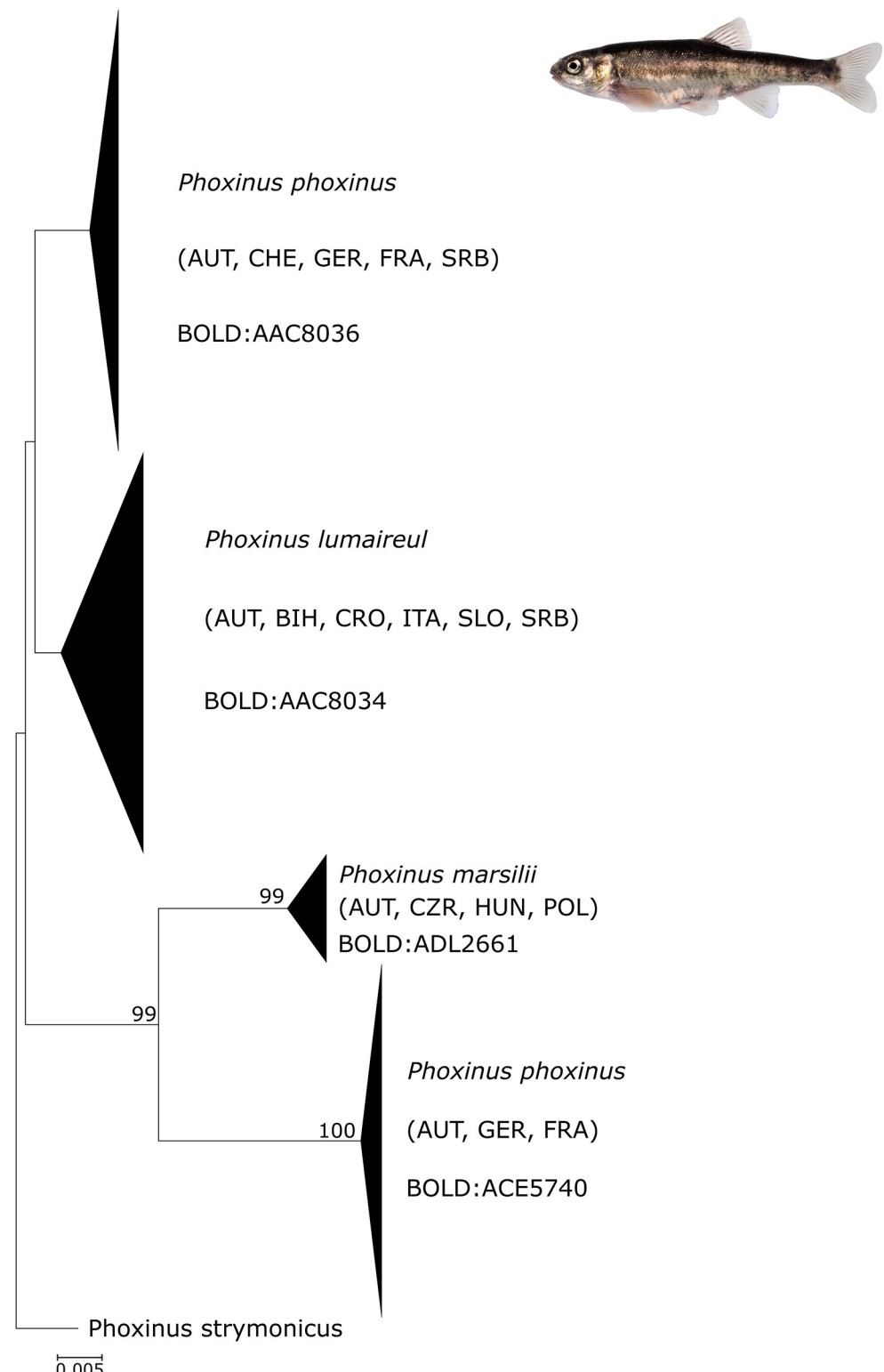

**Fig 5. NJ tree of European *Phoxinus* species.** Phylogeny of European *Phoxinus* species based on *COI* barcode sequences available on BOLD and from this study. Species names and BINs are given, countries of origin are indicated by acronyms in parentheses.

## Taxa sharing BINs

BIN sharing was detected with two species pairs and one trio of species: i) *Leuciscus leuciscus* and *Leuciscus idus*, ii) *Ameiurus nebulosus* and *Ameiurus melas* and iii) *Carassius auratus*, *Carassius gibelio* and *Carassius langsdorfii*. For *L. leuciscus* and *L. idus*, hybridization and mitochondrial replacement has been reported [75], resulting in a shared common haplotype and consequently the same BIN (BOLD:AAD5733). The black bullhead (*Ameiurus melas*) and the brown bullhead (*A. nebulosus*) shared the same BIN (BOLD:AAA7255), even though they are clearly separated in the NJ tree (see Fig 2) and other species delimitation analyses. However, this pattern is not an artefact of the Austrian samples alone, but a general pattern evident on BOLD, as this particular BIN is comprised nearly equally by *A. melas* and *A. nebulosus* samples (https://www.boldsystems.org/index.php/Public_BarcodeCluster?clusteruri=BOLD: AAA7255), underscoring the shallow divergence between the two species. The two species can be clearly distinguished by morphological characters [76], but introgressive hybridization has been reported repeatedly [77 and references therein] and could be an additional problem for molecular delimitation. Furthermore, genetic distances (2.75 DNN) among these two taxa, albeit high enough to support two distinct species, are fairly low compared to most species. Thirdly, the Prussian carp (*Carassius gibelio*) and the goldfish (*Carassius auratus*) share the same BIN with *C. langsdorfii*. All three species belong to the *C. auratus* species complex and have long been considered different sub-species of *C. auratus*, but molecular genetic analyses indicated their distinctness, despite shallow divergence (e.g., [78,79]), a pattern that we also find in our data (see e.g. NJ tree in Fig 2).

## Cases of deep intraspecific divergence

In addition to the few taxa sharing BINs, we found three cases of deep divergence, i.e. in the gudgeons of the genus *Gobio*, in the stone loach, *Barbatula barbatula*, and in the minnows of the genus *Phoxinus*.

Gudgeons of the genus *Gobio* in Austria comprise three distinct mitochondrial lineages that were also resolved as distinct BINs (BOLD:AAC5607, BOLD:ABY6890 and BOLD: ADH1249), which is in sharp contrast to [42,43] who only list one species, *G. gobio*, and [44], who suggest the presence of two species, *G. gobio* and *G. obtusirostris*, for the Austrian Danube system with a potential hybrid zone in the Upper Danube. A recent detailed study [47] found that the three mitochondrial lineages present in Austria correspond to *G. gobio*, *G. obtusirostris* and a third lineage that is closely related to other *Gobio* species from the Balkans. Patterns of genetic diversity suggest that these originally allopatric lineages/species expanded their distribution recently (probably post-glacially) to come into secondary contact and hybridize in the (Austrian) Danube system, thus forming a large hybrid zone in Austria. Even though there seems to be a cline in the relative frequency of the distinct haplogroups from the upper to the lower parts of Danube system [47,80], the distribution of these lineages/species throughout Austria (and adjacent countries) is currently unresolved, and particularly complicated. *Barbatula barbatula* poses another ambiguous case, where sequences from the 17 morphologically identified samples can be allocated into two separate clusters in the NJ tree, forming two BINs (BOLD:AAA1239 and BOLD:AAA1243). This result is partly in line with the three clades recovered by [10], who also found high levels of divergence (<7.02% sequence divergence), potentially indicating cryptic species. The two lineages recovered in Austrian samples (4.66% divergence) are part of the eastern as well as the southern (Danubian) lineage [10] (Fig 4). This pattern also becomes evident when looking at the pan-European dataset (Fig 4). In addition to the Central European lineages, two Eastern/Northeastern lineages were recovered. This finding is consistent with previous studies [10,81], which also found pronounced structure based

on other markers, but did not include Northern European samples. Furthermore, this pattern is similar to what has been observed in gudgeons of the genus *Gobio* [47], with separate glacial refugia and post-glacial secondary contact and admixture. Similarly, additional nuclear genetic or genomic data would be required to comprehensively dis-entangle the complex pattern observed in the mitochondrial data.

The most complex pattern was found in the genus *Phoxinus* (the European minnow species complex). While [44] reported *Phoxinus phoxinus* and *P. lumaireul* for Central Europe, [34,35,46] identified four species and three additional lineages of *Phoxinus* in Austria. These are *Phoxinus marsilii* and *P. lumaireul* (represented by three different subclades), *P. csikii* and *P. phoxinus* (introduced). Discriminating between *Phoxinus* species and dis-entangling their respective distribution ranges and geographical origins is impeded by subtle morphological differences as well as small interspecific genetic variation, which cannot be detected by DNA barcoding. Species delimitation is further complicated by a long and irreproducible history of stocking and translocation as well as hybridization [35]; thus, further in-depth morphological and genetic/genomic investigations are needed.

## First record of ginbuna, *Carassius langsdorfii*, for Austria

Two species of *Carassius*, the Crucian carp (*C. carassius*) and the Prussian carp (*C. gibelio*), are native to Europe. Additionally, the goldfish (*C. auratus*) was introduced in the 17th century as an ornamental fish and has established feral populations throughout Europe (e.g., [44,82,83]), a pattern mirrored by more recent introductions of eastern Asiatic strains of *C. gibelio* [84,85]. Since 2000, another non-native *Carassius* species, C. *langsdorfii*, originally distributed in Japan, has been reported from several European countries [82,86,87], most likely introduced as unintended imports together with koi carps (*Cyprinus rubrofuscus*) [86]. As this species has hitherto not been reported for Austria, our finding of *C. langsdorfii* in the Schwarzaubach in Styria is the first evidence for its occurrence in Austria. Frequent hybridization among *Carassius* species, and between *Carassius* and other cyprinid species, as well as the presence of both sexually reproducing and gynogenetic populations complicate species identification in this genus. In fact, the only species to be reliably identified based on morphology is *C. carassius*, whereas genetic data are indispensable for identifying the other species in the genus (e.g. [82]). Indeed, knowledge about the present distribution of *C. langsdorfii* in Europe is almost exclusively based on mtDNA data [87]. However, a caveat of this strategy is that *Carasissus* species have a high propensity to hybridize, and thus hybridization and introgression might lead to erroneous species identifications when based on mtDNA alone. Nonetheless, the discovery of a *C. langsdorfii* haplotype at least confirms the presence of *C. langsdorfii* mtDNA in Austria. Whether our specimen is indeed *C. langsdorfii* or a hybrid will have to be confirmed by additional, ideally nuclear genetic/genomic data. Phenotypically, this individual has a lower body (with fewer scale rows) than *C. gibelio sensu stricto* caught at the same site (see S1 Fig). The specimen also differed from *C. gibelio sensu stricto* specimens by its lighter ventral and darker dorsal side (compare with [86]), suggesting it might indeed be *C. langsdorfii*.

## Nomenclatural issues

Uncertainties in nomenclature such as in the above-mentioned example of *C. langsdorfii*, but also taxonomic revisions or even 'under-studied' groups constitute an un-negligible issue for online repositories such as BOLD but also museum collections. This becomes apparent when, e.g., looking at gudgeons. Both, [42] and [43] listed *Gobio kesslerii* as present in Austria, whereas [44] already used *Romanogobio kesslerii*. According to [45], however, the correct species name should be *Romanogobio carpathorossicus*, and here we follow this suggestion but

note that *R. carpathorossicus* is listed as a synonym of *R. kessleri* in Eschmeyer's catalogue of fishes [71]. A similar situation is found in gudgeons of the genus *Gobio*, where [42,43] only list *G. gobio*, whereas [44] report *G. gobio* and *G. obtusirostris* from the Danube system with the potential existence of a hybrid zone. The most recent work by [47] however, found three distinct lineages (likely corresponding to *G. gobio*, *G. obtusirostris* and a third, Balkans-derived lineage) to which we also adhere in this study and which was confirmed by [80]. The distribution of these lineages throughout Austria (and adjacent countries) is currently unresolved, and further complicated by high morphological variability and hybridization [47].

Systematics and taxonomy change over time simply due to the accumulation of new or more comprehensive data [45,88–91]. Therefore, museum collections as well as digital (sequence) repositories need to be periodically updated to reflect currently accepted nomenclature. In museum collections, this translates to an iterative additive labelling of physical objects (the verbatim labels are never changed) as well as an immaculate concurrent (digital) documentation [92]. Regarding BOLD, skilled personal observing and incorporating current changes and novelties in the taxonomic backbone are crucial to uphold user confidence and integrity with regards to content. Despite the undisputable requirement of additional effort and resources, this accuracy and timeliness will ensure maximum reliability and use of reference barcode data (in the sense of voucher-related DNA sequences) as well as museum collections for future applications.

This barcode-based inventory of the Austrian fish fauna has brought some new additions [45,47,48,93] and while some of these novelties are shared with adjacent countries [e.g. [10,33], others are original to Austria [45] underscoring the need to update a national Red List. We argue that national red lists should increasingly be augmented by genetic data [10,94–96], which allows for non-invasive monitoring [54] and might illuminate the need for further detailed ecological or systematic study for problematic or ambiguous taxa [31,32]. Here, we provide the first comprehensive DNA barcode reference set for Austrian fishes, which may serve as a basis for a regularly updated Austrian Red List of fish species, aid in sample/specimen identification for both basic and applied monitoring, provide the basis for sound fisheries management and conservation of native fish populations and facilitate read determination in eDNA or meta-barcoding studies. Furthermore, our data update helps to increase the coverage of barcoding data at the European scale and thus will likely be useful in a wider biogeographic context.

## Supporting information

**S1 Fig. Pictures of *Carassius* samples from Schwarzaubach, Styria.** A) *C. langsdorfii*, B-D) *C. gibelio*.
(TIF)

**S1 Table. Summary of species delimitation analyses results.**
(DOCX)

## Acknowledgments

We are grateful to Günter Parthl, Gerhard Woschitz, Albert Rechberger, Josef Melcher, Peter Mehlmauer, Harald Ellinger, Michael Jung, Clemens Ratschan, Klemens Koblmüller, Mario Poglitsch and the Fischereiverein Leibnitz for their help acquiring various samples during electric fishing surveys. We also thank Taraneh E. H. Westergerling, Raphael Schmid and Elisabeth Glatzhofer for their help with the laboratory work. We also kindly acknowledge Sandra Kirchner and Oliver Macek from the Natural History Museum in Vienna for managing the

reference specimens and providing Museum IDs and for help with database work, respectively. We would also like to thank Anja Palandačić for her comments and suggestions on the topic of minnows in Austria.

## Author Contributions

**Conceptualization:** Lukas Zangl, Steven J. Weiss, Stephan Koblmüller.

**Data curation:** Lukas Zangl, Sylvia Schäffer, Daniel Daill, Thomas Friedrich, Wolfgang Gessl, Marija Mladinić, Josef Wanzenböck, Steven J. Weiss, Stephan Koblmüller.

**Formal analysis:** Lukas Zangl, Thomas Friedrich.

**Funding acquisition:** Christian Sturmbauer, Stephan Koblmüller.

**Investigation:** Lukas Zangl, Josef Wanzenböck, Steven J. Weiss.

**Methodology:** Lukas Zangl, Sylvia Schäffer, Daniel Daill, Stephan Koblmüller.

**Project administration:** Lukas Zangl, Daniel Daill, Christian Sturmbauer, Josef Wanzenböck, Steven J. Weiss, Stephan Koblmüller.

**Resources:** Christian Sturmbauer, Stephan Koblmüller.

**Software:** Lukas Zangl, Stephan Koblmüller.

**Supervision:** Lukas Zangl, Christian Sturmbauer, Steven J. Weiss, Stephan Koblmüller.

**Validation:** Lukas Zangl, Sylvia Schäffer, Daniel Daill, Thomas Friedrich, Wolfgang Gessl, Marija Mladinić, Christian Sturmbauer, Josef Wanzenböck, Steven J. Weiss, Stephan Koblmüller.

**Visualization:** Lukas Zangl, Wolfgang Gessl, Stephan Koblmüller.

**Writing – original draft:** Lukas Zangl, Sylvia Schäffer, Daniel Daill, Thomas Friedrich, Wolfgang Gessl, Marija Mladinić, Christian Sturmbauer, Josef Wanzenböck, Steven J. Weiss, Stephan Koblmüller.

**Writing – review & editing:** Lukas Zangl, Sylvia Schäffer, Daniel Daill, Thomas Friedrich, Wolfgang Gessl, Marija Mladinić, Christian Sturmbauer, Josef Wanzenböck, Steven J. Weiss, Stephan Koblmüller.

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
