## [Decision Letter · Decision Letter 0]

1 Mar 2022

PONE-D-21-40952A comprehensive DNA barcode inventory of Austria’s fish and lamprey speciesPLOS ONE

Dear Dr. Zangl,

Thank you for submitting your manuscript to PLOS ONE. After careful consideration, we feel that it has merit but does not fully meet PLOS ONE’s publication criteria as it currently stands. Therefore, we invite you to submit a revised version of the manuscript that addresses the points raised during the review process. 1) Both reviewers suggest a number of text changes to improve the clarity of the text. These should be included in a revised version of the manuscript.  2) Please do not forget to provide the accession numbers for the newly described sequences.  

We look forward to receiving your revised manuscript.

Kind regards,

Sebastian D. Fugmann, Ph.D.

Academic Editor

PLOS ONE

Journal Requirements:

4. Please include a copy of Table 1 which you refer to in your text on page 9.

Reviewers' comments:

Reviewer's Responses to Questions

**Comments to the Author**

1. Is the manuscript technically sound, and do the data support the conclusions?

Reviewer #1: Yes

Reviewer #2: Yes

2. Has the statistical analysis been performed appropriately and rigorously? 

Reviewer #1: Yes

Reviewer #2: N/A

3. Have the authors made all data underlying the findings in their manuscript fully available?

Reviewer #1: Yes

Reviewer #2: Yes

4. Is the manuscript presented in an intelligible fashion and written in standard English?

Reviewer #1: Yes

Reviewer #2: Yes

5. Review Comments to the Author

Reviewer #1: In this study, the authors used DNA barcoding to explore the fish diversity in Austrian waters. A total of 1014 barcodes, of which 333 sequences recovered from BOLD, have been yielded representing 96% of all species present in Austria. Furthermore, the authors support their results by using three different species delimitation methods. I suggest publishing the paper after some revisions.

Abstract: line 32: should also Gobio sp be referred here?

48: “determine different ontogenetic stages”: this phrase is not clear. DNA barcoding cannot determine different ontogenetic stages but can identify species regardless of the ontogenetic stage. Please rephrase.

Materials and methods:

Authors firstly present the dataset they used: how these sequences were produced? You should rearrange the materials and methods starting from the collection of the samples

154-155: in “materials and methods” section authors refer that 681 barcodes whereas here this number is 740. Which is the correct one? Additionally, how the overall success rate of 93% is calculated?

164-173: This part is a bit confusing, especially the last sentence. In the first sentence you are referred to multiple BINS for one species (or genus), then to BIN sharing and in the last one to multiple BINS for one species (again). I assume that as regards multiple BINS in the first case you mean the BINS which are produced by your results whereas in the second case one BIN comes from the present study and the second one from BOLD data (considering also the Table S1). Please explain better what you mean.

199: “North Macedonia” instead of “Makedonia”

215: why missing? Authors may explain why these species are missing

215-219: Would 1-2 individuals be too little to represent a species? Can we ensure the accuracy of the morphological identification on-site?

Finally, two more general comments are presented:

1. It would be useful to present the number of individuals per species (only for a few species it is clear in the manuscript) (eg in a table).

2. Authors may include a map including all the collection sites.

3. Please provide Genbank accession numbers and BOLD IDs for the sequences produced by the present study (I assume that you have already submitted them).

Reviewer #2: This manuscript is about a barcode database from fish biodiversity inventory from Austria.

Despite the barcode method has been a common tool to access the biodiversity, this manuscript brings an important update for the list of fish species from Austria. Furthermore, also confirm some previous information about new lineages occurring in Europe. It is an important DNA repository for other studies and also as reference for identifications.

The manuscript increases the molecular database for Europe, but it should be improved the English language, as well as revise some concepts used in the wrong definition (noted below).

The introduction is most focused on the method of Barcode and how this tool is effective to studies in general. However, I missed more information about Austria fish fauna, the last assessment, and the biodiversity known. So, the reader will be able to compare with the results of the present study.

I also did not see any legend for the figures (except by fig.2), as well as for the tables.

I general, the manuscript has a merit in terms of quantity and important data, but it has to improve many things before be published.

Follow below some corrections and suggestions to authors. I hope this can help to improve the text.

Best wishes!

Title: remove “lampreys” from the title since they are also fish. I do not understand why authors mentioned as a separate group. So, change the title to “A comprehensive DNA barcode inventory of Austria’s fish species”

Line 20-21: once more why lampreys are treated as separate from fish since it is also fish?

Line 21: change “Red List” for “Red List of threatened species”

Line 22: delete “lamprey species”

Line 24: change “inventory” for “database”; delete “lamprey species”

Line 25: change “COI” for “coI”. There is an agreement to standardize the abbreviation of gene names, and particularly for fish, in which all codified genes for fish groups should be abbreviated in lower case and italic.

Line 27: delete “almost perfectly”

Line 28: what do you mean in this case with morphospecies? Since the analysis resulted in undescribed species, this new morphotypes were already known to science? I think only “species” is more suitable here and then you can add new forms for the probably new species or distinct lineages.

Line 48: I am not sure if this is possible, as I did not read the paper mentioned. But, I think maybe you want to mean “identify juveniles and larvae rather than determine different ontogenetic stages. The molecular is equal for adults or juveniles, so I did not get it. Please revise this.

Line 106: change for “coI sequences of Austrian fish species already available from BOLD…”

Line 107: space between 681 and new

Line 108-112: You do not need inform the total number of sequences before you remove the unidentified species or shorter ones. Present the number used in the analysis (i.e. 1,014) and remain the observation that you excluded the unidentified species and shorter sequences from the analysis. The number of thousand should be present as 1,014.

Line 113: change “non-Austrian sequences from elsewhere in Europe” for “sequences from other region of Europe, outside of Austria, were used from BOLD…”

Line 119: which Museum? Different museum ? listed on…?

125: which were the parameters for the amplification of coI? It should be written here.

Line 154: How could be 740 barcode sequences if you previously mentioned 742?

Line 155: overall success rate of what? Of identification at species level? Something is missing here. I also think that present the covering data in number not in percentage is more reliable since we can have a better ideia about it. Or, as the covering is high perhaps mention how species the study did not contemplate.

Line 158; there isn`t “well resolved” in phylogenies and tree. It is resolved or not. Perhaps you meant “well supported”. My suggestion is delete “well” from the sentence.

Line159: Also, there is not “perfectly” in terms of taxonomy, so delete this word from the text, and change for “the clades match with morphological species identification”.

Line 163: this example of C. langsdorfii is was not a discordance between DNA and morphospecies, actually this is not the conceptual of morphospecies. This was an clear example of a misidentification.

Line 164: change morphospecies for morphotypes. Morphospecies is other conceptual and need very different analysis to show the space of the shape of the species. Not about different phenotypes.

Line 177: This could be also a case of molecular contamination sample during Lab techniques.

Line 177 and 178: What do you mean with Lmax? It is not mentioned in M&M

Line 180-183: sort species in alphabetic order.

Line 191: cite for the first time the entire name of the genus

Line 315 and 316: There is not “true” for the species. Change for stricto sensu

Line 323: What do you mean with “physical” museum collection?

Line 324 and 325: The phrase is understandable.

Line 325-326: This is not an example of a issue, since the species is the same, it is matter of with genus is included the species. Doesn’t change the ID of the species.

Line 328:This is not about to decide between the use of the name R. kessleri (or G. kessleri) or R. carpathorossicus. It is about if R. carpathorossicus is a valid species or not. If not (in case of junior synonym of R. kessleri), thus the it should use the prevalence rule of the oldest name, i.e. R. kessleri.

Line 339: the repositories is not for reflect the current accept nomenclature but rather for support the nomenclature decisions.

Figures: except by Fig. 2, they don`t have legend, as well as the supplementary tables.

6. PLOS authors have the option to publish the peer review history of their article (what does this mean?). If published, this will include your full peer review and any attached files.

Reviewer #1: No

Reviewer #2: No

---

## [Author Response · Author response to Decision Letter 0]

29 Mar 2022

All comments from the reviewers and editor have been addressed and can be checked in the 'Response to reviewers' file.

---

## [Decision Letter · Decision Letter 1]

6 May 2022

A comprehensive DNA barcode inventory of Austria’s fish species

PONE-D-21-40952R1

Dear Dr. Zangl,

We’re pleased to inform you that your manuscript has been judged scientifically suitable for publication and will be formally accepted for publication once it meets all outstanding technical requirements.

Kind regards,

Sebastian D. Fugmann, Ph.D.

Academic Editor

PLOS ONE

Additional Editor Comments (optional):

Reviewers' comments:

Reviewer's Responses to Questions

**Comments to the Author**

1. If the authors have adequately addressed your comments raised in a previous round of review and you feel that this manuscript is now acceptable for publication, you may indicate that here to bypass the “Comments to the Author” section, enter your conflict of interest statement in the “Confidential to Editor” section, and submit your "Accept" recommendation.

Reviewer #1: All comments have been addressed

2. Is the manuscript technically sound, and do the data support the conclusions?

Reviewer #1: Yes

3. Has the statistical analysis been performed appropriately and rigorously? 

Reviewer #1: Yes

4. Have the authors made all data underlying the findings in their manuscript fully available?

Reviewer #1: Yes

5. Is the manuscript presented in an intelligible fashion and written in standard English?

Reviewer #1: Yes

6. Review Comments to the Author

Reviewer #1: Figure S2 is missing and authors should provide it.

Except of this, all comments have been addressed and I suggest the publication of the paper.

7. PLOS authors have the option to publish the peer review history of their article (what does this mean?). If published, this will include your full peer review and any attached files.

Reviewer #1: No

---

## [Editor Report · Acceptance letter]

23 May 2022

PONE-D-21-40952R1 

A comprehensive DNA barcode inventory of Austria’s fish species 

Dear Dr. Zangl:

I'm pleased to inform you that your manuscript has been deemed suitable for publication in PLOS ONE. Congratulations! Your manuscript is now with our production department. 

Kind regards, 

on behalf of

Dr. Sebastian D. Fugmann 

Academic Editor

PLOS ONE